# Deletion of Thioredoxin-Interacting Protein (TXNIP) Abrogates High Fat Diet-Induced Retinal Leukostasis, Barrier Dysfunction and Microvascular Degeneration in a Mouse Obesity Model

**DOI:** 10.3390/ijms21113983

**Published:** 2020-06-01

**Authors:** Islam N. Mohamed, Nader Sheibani, Azza B. El-Remessy

**Affiliations:** 1Department of Pharmaceutical and Biomedical Sciences, College of Pharmacy, California North State University, Elk Grove, CA 95758, USA; islam.mohamed@cnsu.edu; 2Research Service Line, Charlie Norwood VA Medical Center, Augusta, GA 30912, USA; 3Departments of Ophthalmology and Visual Sciences, Cell and Regenerative Biology, and Biomedical Engineering, University of Wisconsin School of Medicine and Public Health, Madison, WI 53705, USA; nsheibanikar@wisc.edu; 4Department of Surgery, 820 South Wood Street, University of Illinois at Chicago, Chicago, IL 60612, USA

**Keywords:** obesity, high fat diet, leukostasis, inflammasome, inflammation, retina, acellular capillaries, vascular permeability, TXNIP, NLRP3, caspase-1, IL-1β

## Abstract

We have shown that a high fat diet (HFD) induces the activation of retinal NOD-like receptor protein (NLRP3)-inflammasome that is associated with enhanced expression and interaction with thioredoxin-interacting protein (TXNIP). Here, the specific contribution of TXNIP and the impact of HFD on retinal leukostasis, barrier dysfunction and microvascular degeneration were investigated. Wild-type (WT) and TXNIP knockout (TKO) mice were fed with normal diet or 60% HFD for 8–18 weeks. TXNIP was overexpressed or silenced in human retinal endothelial cells (REC). At 8 weeks, HFD significantly induced retinal leukostasis and breakdown of the blood–retina barrier in WT mice, but not in TKO mice. In parallel, HFD also induced retinal expression of adhesion molecules and cleaved IL-1β in WT mice, which were also abrogated in TKO mice. In culture, TXNIP overexpression induced NLRP3, IL-1β, and adhesion molecules expression, while TXNIP silencing inhibited them. Blocking the IL-1β receptor significantly suppressed TXNIP-induced expression of NLRP3-inflammasome and adhesion molecules in HREC. Ex-vivo assay showed that leukocytes isolated from WT-HFD, but not from TKO-HFD, induced leukostasis and cell death. At 18 weeks, HFD triggered development of degenerated (acellular) capillaries and decreased branching density in WT but not in TKO mice. Together, HFD-induced obesity triggered early retinal leukostasis and microvascular dysfunction at least in part via TXNIP-NLRP3-inflammasome activation.

## 1. Introduction

Retinal inflammation and leukostasis, defined by abnormal adhesion between activated leukocytes and microvasculature, are well-defined milestones in the development of diabetic retinopathy (DR) (reviewed in [1,2]). Further, leukocytes can uniquely target injured tissues, adhere to the vasculature, and locally secrete pro-inflammatory mediators. These events are paralleled by increased expression of intracellular adhesion molecules, which accelerate the onset of leukostasis [1]. The consequent release of cytokines such as tumor necrosis factor-alpha (TNF-α) and interleukin 1-beta (IL-1β) can directly result in blood retinal barrier (BRB) breakdown. Retinal inflammation has been closely linked to exacerbated loss of retinal endothelial cells (REC) and formation of the non-perfused degenerated acellular capillaries, a hallmark of ischemia [3,4]. A growing body of evidence has recognized sterile inflammation manifested by increased retinal caspase-1 activation and IL-1β levels to have a causative role in initiating and sustaining retinal microvascular dysfunction and degeneration [5,6,7]. Interestingly, the majority of the evidence linking inflammatory processes to development of DR has extensively focused on insulin-deficient rather than insulin-resistant rodent models.

Obesity has been recently upgraded to a disease state rather than a mere risk factor for developing dyslipidemia and insulin resistance, which culminate in the complex metabolic syndrome disorder and type-2 diabetes [8]. Several studies have characterized each of these metabolic conditions as an independent risk factor for retinal microvascular abnormalities in non-diabetic [9,10,11,12] or diabetic populations [12], leading eventually to the development of DR. Metabolic-associated systemic low-grade inflammation, oxidative stress and REC activation and dysfunction have been suggested as the most plausible reasons [8,13,14]. Nevertheless, little is known about the molecular mechanisms involved in instigating obesity-associated insulin-resistance and inflammation on retinal microvascular integrity and function.

A high fat diet (HFD) was reportedly shown to induce the expression of thioredoxin-interacting protein (TXNIP) in various models [15,16,17,18]. TXNIP is the endogenous inhibitor of thioredoxin, the ubiquitous antioxidant defense protein system, and is known for its pro-inflammatory, pro-oxidative stress and pro-apoptotic activity (reviewed in [19]). TXNIP also acts as a direct activator of the NOD-like receptor protein (NLRP3)-inflammasome in different cell types [15,18,20,21,22,23]. The NLRP3-inflammasome is one of the most established multi-protein complexes known for instigating obesity-induced inflammation [24]. Activated NLRP3 oligomerizes with the ASC (apoptosis-associated speck-like) adaptor protein which recruits procaspase-1, allowing its autocleavage and activation, which in turn cleaves IL-1β into active mature form before its release [19,25]. Our group previously demonstrated that TXNIP is required for the activation of the upstream signaling complex, the NLRP3-inflammasome and resulting inflammation in liver and skeletal muscle from HFD-fed mice [20,26]. Here, we examined the impact of HFD as well as TXNIP deletion on retinal leukostasis, barrier dysfunction and microvascular degeneration in vivo. In addition, the relative contribution of TXNIP in REC cultures versus leukocytes has been dissected using ex-vivo assays of REC culture and isolated peripheral mononuclear blood cells from wild and TXNIP-deficient mouse models. To the best of our knowledge, this is the first report showing retinal leukostasis, barrier dysfunction and accelerated microvascular degeneration in a HFD-induced insulin resistant model. These studies support the notion that targeting TXNIP expression provides a potential therapeutic tool to protect against HFD-associated retinal early microvascular dysfunction.

## 2. Results

### 2.1. HFD Impact on TXNIP Expression and Metabolic Profiles in WT and TKO Mice

Six-week-old age and gender matched C57Bl/6J wild-type mice (WT) and TXNIP knockout (TKO) mice were inbred at the facility and randomized for feeding with either standard chow (normal diet; WT-ND and TKO-ND groups) or high fat diet (WT-HFD and TKO-HFD groups) for 8 weeks or 18 weeks. We confirmed the impact of HFD to trigger retinal TXNIP expression in mice similar to our previous findings in rats [16]. As shown in Appendix A, 8 weeks of HFD triggered a 1.7-fold increase in retinal TXNIP expression in WT compared to normal diet controls, while TKO showed minimal expression (* *p* < 0.05, *n* = 4–6). Mice were weighed weekly and clearly HFD resulted in similar increases in weight gain in both WT-HFD and TKO-HFD mice when compared to their ND-controls over the 18 weeks (* *p* < 0.05, *n* = 11, Figure 1a). However, area under the curve (AUC) analysis indicated that WT-HFD gained overall more weight when compared to TKO-HFD (# *p* < 0.05, *n* = 11, Figure 1b). Fasting blood glucose levels were recorded at 8, 12 and 18 weeks, and HFD induced a modest yet significant increase in fasting blood glucose levels in WT-HFD when compared with the WT-ND group at 8 weeks through 18 weeks of study (* *p* < 0.05, *n* = 11–19, Table 1). TKO mice showed lower blood glucose levels and were partially protected against HFD-induced insulin resistance observed in WT-HFD mice as described before [26].

### 2.2. Deletion of TXNIP Abrogates HFD-Induced Retinal Leukostasis and BRB Breakdown

Retinal leukostasis was assessed by counting the number of adherent leukocytes labelled with Concanavalin-A that were physically occluding the poorly perfused retinal microvessels (Figure 2a). After 8 weeks, HFD significantly increased the number of adherent leukocytes (2.4-fold) in the retinal capillaries from WT-HFD when compared with WT-ND (Figure 2b, *p* < 0.05, *n* = 8–11). Deletion of TXNIP significantly prevented leukostasis in TKO-HFD compared with WT-ND and TKO-ND groups. Next, we evaluated the effects of HFD on inducing retinal vascular injury and BRB breakdown. As shown in Figure 2c, HFD induced BRB dysfunction evident by a 2.5-fold increase in extravasation of BSA-fluorescence in WT-HFD. Deletion of TXNIP preserved BRB function against HFD-mediated barrier dysfunction when compared to TKO-ND (*p* < 0.05, *n* = 5–8). 

### 2.3. Deletion of TXNIP Prevents HFD-Induced Retinal Inflammation and Expression of Cell Adhesion Molecules

Traditionally, leukostasis is provoked by expression of endothelial cell adhesion molecules. Expression of intercellular adhesion molecules-1 (ICAM-1) and vascular cell adhesion molecule-1 (VCAM-1) was assessed after 8 weeks of HFD. As shown in Figure 3a,b, HFD significantly increased the expression of ICAM-1 (1.5-fold) and VCAM-1 (2.3-fold) in WT-HFD compared with WT-ND (*p* < 0.05, *n* = 4–6). Deletion of TXNIP significantly mitigated the HFD-induced expression of adhesion molecules (*p* < 0.05, *n* = 4–6).

### 2.4. Deletion of TXNIP Abrogates HFD-Induced Activation of NLRP3-Inflammasome in the Retina

Our group previously showed that HFD increases the activation of NLRP3-inflammasome and drives the association between TXNIP and NLRP3 in the retina [16,17]. Here we examined the impact of HFD and TXNIP deletion on expression of NLRP3, cleaved caspase-1 and IL-1β after 8 weeks. HFD resulted in a significant increase (3-fold, *p* < 0.05, *n* = 10–12) in the retinal cleaved IL-1β expression in WT-HFD mice compared with WT-ND group (Figure 4a,c). HFD did show a trend but no significant increases in retinal NLRP3 expression (Appendix A) or cleaved caspase-1 (Figure 4a,b) when compared with ND (*n* = 10–12). TXNIP deletion abolished the expression of NLRP3 (Appendix A), cleaved caspase-1 (Figure 4a,b) and cleaved IL-1β (Figure 4a,c) in both TKO-ND and TKO-HFD mice compared with the WT-ND group.

### 2.5. Overexpression of TXNIP Activates NLRP3-Inflammasome in an IL-1β Dependent Fashion

The results from whole retina studies suggested that TXNIP is required for increased expression of IL-1β and activation of the NLRP3 inflammasome. To examine the causal role of TXNIP in HFD-induced endothelial inflammation, TXNIP was overexpressed using electroporation of TXNIP-plasmid in human REC. As shown in Figure 5a–c, overexpression of TXNIP (TXNIP ^+ +^) induced the upregulation of NLRP3 expression (2.5-fold) and a trend towards increased expression of cleaved caspase-1 (1.4-fold) compared with Empty vector + GFP (EV-GFP)-controls in REC (*p* < 0.05, *n* = −6). As shown in Figure 5d, overexpression of TXNIP (TXNIP ^+ +^) induced the activation of the NLRP3-inflammasome as evident by cleaved IL-1β expression (2.4-fold) in REC lysates. This was paralleled with a similar increase in endothelial inflammation evident by increased TNF-α expression by 1.85-fold in the TXNIP ^+ +^ group compared with EV-GFP-controls (Figure 5e,f, *p* < 0.05, *n* = 5–6). Since IL-1β has a critical role in mediating retinal microvascular dysfunction (reviewed in [19,27]), we examined the effect of neutralizing IL-1β receptor signaling in REC. Treatment with IL-1R antagonist suppressed the effect of TXNIP overexpression not only on inflammasome activation and activation of IL-1β, but also on the TNF-α expression. Of note, treatment with IL-1R antagonist tended to increase TXNIP expression in GFP controls and reduced TXNIP expression in TXNIP + + and but did not reach statistical significance (Appendix A). These results highlight an essential role for IL-1β in TXNIP-mediated inflammation in an autocrine fashion in REC.

### 2.6. Silencing of TXNIP Abolishes While Its Overexpression Induces EC Adhesion Molecule Upregulation in an IL-1β Dependent Fashion

To further investigate the causal role of TXNIP in HFD-induced microvascular inflammation, TXNIP expression was silenced in human REC using siRNA, which were then stimulated with saturated fatty acid “palmitate” coupled to BSA (Pal-BSA) as described previously [16,17]. Silencing TXNIP expression using siRNA was confirmed in human REC cultures which showed a 3-fold increase in TXNIP expression in response to Pal-BSA, compared to BSA control treatment in cells transduced with scrambled RNA. Transduction with siRNA against TXNIP significantly abrogated TXNIP expression when compared to cells transduced with scrambled RNA (Appendix A; *p*-value < 0.05, *n* = 3). As shown in Figure 6a–c, Pal-BSA also resulted in significant increases in expression of ICAM-1 (1.7-fold) and platelet endothelial cell adhesion molecule (PECAM-1) (1.5-fold) compared with BSA controls (*p* < 0.05, *n* = 3–4). Silencing TXNIP expression abolished such responses in REC. Next, we evaluated the direct role of TXNIP overexpression and neutralizing IL-1β receptor signaling in modulating the expression of adhesion molecules. As shown in Figure 6d–f, overexpression of the TXNIP plasmid (TXNIP ^++^) significantly induced the expression of adhesion molecules ICAM-1 (1.5-fold) and PECAM-1 (1.65-fold) compared to the empty vector (EV-GFP) control group (*p* < 0.05, *n* =5–6). Treatment of human REC with IL-1R inhibitor blocked the enhanced expression of adhesion molecules in TXNIP-overexpressing cells (Figure 6d–f).

### 2.7. Deletion of TXNIP in Leukocytes Prevents Ex-Vivo Leukostasis and REC Death

In order to examine the role of leukocyte-TXNIP in retinal leukostasis, isolated peripheral blood mononuclear cells (PBMNCs) from all mice groups (after 18 weeks of HFD) were incubated with mouse REC ex-vivo. PBMCs isolated from the WT-HFD group showed a 2-fold increase in the number of adherent labeled PBMCs compared with the WT-ND control group (Figure 7a,b, *p* < 0.05, *n* = 5–6). In parallel, PBMCs isolated from WT-HFD group showed a 1.8-fold increase, yet did not reach statistical significance in TUNEL-positive nuclei compared with the WT-ND control group (Figure 7c, *p* = 0.06, *n* = 5–6). In contrast, PBMCs isolated from TKO-HFD or TKO-ND showed no marked changes in REC death compared with WT-ND control group (Figure 7c).

### 2.8. Deletion of TXNIP Abrogates HFD-Induced Retinal Microvascular Degeneration

Finally, we evaluated the effects of HFD on inducing retinal microvascular degeneration assessed by development of occluded acellular capillaries, the hallmark of retinal ischemia, after 18 weeks. HFD resulted in a significant increase (1.5-fold) in acellular capillary formation in WT-HFD but not in TKO-HFD compared with WT-ND (Figure 8a,b, *p* < 0.05, *n* = 6). This was associated with a 1.8-fold increase towards upregulation of pro-apoptotic protein cleaved PARP expression in the WT-HFD group compared with the WT-ND control group (Figure 8d,e, *p* = 0.1, *n* = 6). Interestingly, deletion of TXNIP abolished the increase in cleaved PARP when compared to WT-HFD.

### 2.9. Deletion of TXNIP Abrogates HFD-Associated Retinal Microvascular Morphological Changes

In parallel, HFD also resulted in decreased retinal vascular branching density by 30% compared with the WT-ND control group. In contrast, TKO mice had comparable vessel branching density compared with WT-ND, which did not show changes in response to HFD (Figure 9b). We have also observed a similar trend in the overall vascular density per retina area (*p* > 0.05), but no change in vascular tortuosity was observed among all groups (data not shown).

## 3. Discussion

In this study, we investigated the protective effect of TXNIP deletion against HFD-induced retinal NLRP3-inflammasome activation and its associated microvascular injury. We also attempted to dissect the underlying molecular mechanisms governing the interaction between observed landmarks of early stage pre-diabetic retinopathy. The major findings of our study are: (1) TXNIP is required for HFD-induced NLRP3-inflammasome activation and retinal vascular injury; (2) HFD resulted in TXNIP-dependent retinal microvascular inflammation and adhesion molecule expression that was associated with increased leukostasis, BRB breakdown and microvascular degeneration; (3) Silencing of TXNIP abolished retinal EC inflammation and adhesion molecule upregulation in an IL-1β autocrine positive-feed forward loop fashion, while its overexpression induced this; and (4) HFD may contribute to EC death through circulating leukocytes, which are also inhibited upon TXNIP deletion.

Retinal leukostasis is one of the established indirect prerequisite events for inducing exacerbated EC death and BRB breakdown, and its inhibition can help to prevent retinal acellular capillary formation [4,28,29]. Our results indicated that HFD induced NLRP3-inflammasome activation, which was associated with parallel increases in retinal vascular cell adhesion molecules, leukostasis and BRB breakdown after 8 weeks in WT-HFD compared with ND-controls (Figure 2, Figure 3 and Figure 4). These results concurred with previous reports that showed significant increases in ICAM-1 expression after 8 weeks of the HFD model [30,31]. Our studies lend further credit to a previous report showing increased retinal leukostasis in the non-diabetic fatty Zucker insulin-resistant rats [32], and a later study that reported blood barrier dysfunction after 16 weeks of HFD [33]. Further, a recent study showed that retinas from HFD mice manifested striking induction of stress kinase and neural inflammasome activation at 3 months, before the development of electroretinographic defects or microvascular disease [34].

TXNIP, a member of the α-arrestin family of adaptor and scaffolding proteins [19], activates the canonical NFκB and its downstream wide array of pro-inflammatory cytokines including IL-1β, ICAM-1, TNF-α, VEGF-A and Cox-2 expression [35,36,37,38,39]. Deletion of TXNIP prevented HFD-induced adhesion molecules and leukostasis in vivo, and silencing TXNIP abolished palmitate-induced upregulation of adhesion molecules ICAM-1 and PECAM-1 in vitro (Figure 4). Moreover, enhanced expression of TXNIP in EC resulted in activation of the NLRP3-inflammasome evidenced by increased NLRP3 and cleaved IL-β expression, along with TNF-α and both adhesion molecules ICAM-1 and PECAM-1, respectively. Interestingly, blocking the effect of IL-1β by the IL-1RA was able to not only suppress the induction of its own processing pathway of NLRP3-inflammasome, but also the expression of the other pro-inflammatory mediators, including TNF-α, ICAM-1 and PECAM-1 adhesion molecules. These findings establish the pro-inflammatory role of TXNIP as a direct activator of pro-inflammatory adhesion molecule expression in EC, and possibly via NLRP3-inflammasome activation and IL-1β expression in an autocrine positive-feed forward loop fashion. In agreement, increased and auto-sustained IL-1β production has been previously demonstrated in retinal EC and macroglia in diabetic models [40]. Therefore, it is reasonable to consider the contribution of other metabolic pathways including dyslipidemia or altered insulin sensitivity that might be affected by TXNIP deletion.

Retinal leukostasis is one of the established indirect prerequisite events for inducing exacerbated EC death and BRB breakdown, and its inhibition prevents retinal acellular capillary formation [4,28,29]. In agreement, our results showed that HFD-induced leukostasis after 8 weeks was positively correlated with development of acellular capillaries and decreased vascular density after 18 weeks in WT-HFD, but not in TKO mice. Interestingly, a combination of HFD and STZ for simulation of type 2 diabetes accelerated development of retinal acellular capillaries and barrier dysfunction [41,42]. Inhibition of caspase-1 or deletion of IL-1 receptor prevented IL-1β-dependent formation of acellular capillaries in retinas of type 1 diabetic mouse models [27]. Our findings establish the detrimental long-term impact of HFD alone on retinal microvascular degeneration and morphological abnormalities, and highlight the protective effect of TXNIP deletion. Furthermore, 18 weeks of HFD also induced increased formation of retinal acellular capillaries that were associated with a trend in increased levels of the pro-apoptotic marker cleaved PARP in WT-mice but not in TKO-mice (Figure 7). Clinically, retinal microvascular capillary obliteration and degeneration can be evident as non-perfused areas in fluorescein-infusion retinal angiograms [2,28]. In line with the observed increases in retinal microvascular degeneration, WT-HFD mice also showed decreased branching density that was absent in TKO mice. Retinal microvascular cell death can occur either directly due to different biochemical insults initiated within retinal EC themselves, or indirectly secondary to the activation of other retinal cell types including neurons and glial cells, or non-retinal cell types, mainly circulating or infiltrating leukocytes [43]. Previously, we showed that TXNIP was required for the direct palmitate-induced NLRP3-inflammasome activation and its associated EC death [16]. Here we examined the interaction between isolated PBMCs from all animal groups. Isolated PBMCs from WT-HFD mice but not TKO-HFD mice resulted in increased leukostasis and apoptosis (*p* = 0.07 compared with WT-ND) of mouse retinal EC cultures, after 2 and 24 h in co-culture, respectively. Our recent studies using the HFD model showed significant increases in IL-1β serum level that was mitigated by genetic deletion of TXNIP [26]. Together, these results highlight a potential role for circulating leukocytes in mediating EC death in models of HFD via both direct and indirect EC death pathways.

Further studies are warranted to dissect the role of hyperglycemia versus saturated fatty acid and altered insulin signaling in diabetic versus obese pre-diabetic or insulin-resistant conditions. Such interactions can contribute to the early pathological manifestation of pre-diabetic retinopathy and retinal microvascular abnormalities observed in patients with obesity and metabolic syndromes, independently from or in addition to diabetes [11,12]. In the era when obesity has been upgraded to an independent disease state, rather than a mere risk factor, such paradigm shifts suggest the TXNIP-NLRP3-inflammasome axis as a promising therapeutic target for early treatment or prevention of obesity-associated microvascular diseases, benefiting more than 80 million Americans.

## 4. Materials and Methods

### 4.1. Animal Preparation and Metabolic Profile

All animal studies were in accordance with the Association for Research in Vision and Ophthalmology and the Institutional Animal Care and Use Committee at the Charlie Norwood VA Medical Center (ACORP no. 15–04–080), 24 April 2015. Six-week-old age and gender matched C57Bl/6J wild-type (WT) mice and TXNIP knock out (TKO) mice were inbred at the facility and randomized for feeding with either standard chow (normal diet; WT-ND and TKO-ND groups) or high fat diet (60% fat: Research diets, Product #D12492; WT-HFD and TKO-HFD groups) for 8 weeks or 18 weeks. Mice were weighed weekly and blood glucose levels were determined at 8, 12 and 18 weeks (Table 1 and Figure 1).

### 4.2. Determination of Blood–Retina Barrier Function

Integrity of the BRB was evaluated as described previously [39]. Mice received jugular vein injections of 10 mg/kg BSA-conjugated fluorescein (Invitrogen). After 20 min, animals were sacrificed, and blood and retinas were collected. Retinas were homogenized in RIPA-buffer to detect their fluorescence (excitation 370 nm, emission 460 nm) using a plate reader (BioTek Synergy2, Winooski, VT, USA). Retina fluorescence was normalized to that of serum and to total protein content of each retina sample.

### 4.3. Leukostasis 

*Leukostasis* was assessed by “concanavalin A” labeling and quantification of adherent leukocytes as previously described [44]. Briefly, mice were transcardially perfused with fluorescent isothiocyanate (FITC)-labeled concanavalinA (ConA) lectin (Vector Laboratories, Burlingame, CA). Whole eyes were fixed with 4% paraformaldehyde overnight and retinal flat-mounts were then imaged. Adherent leukocytes were counted from at least 3–6 different fields of the mid-retinal area and calculated as the average number/field using an AxioObserver.Z1 Microscope (Zeiss, Germany). The ConA-positive blood cells were identified as leukocytes based on their size (7–20 µm diameter), morphology (round or oval shape) and intraluminal position clogging the retinal microvasculature.

### 4.4. Isolation of Peripheral Blood Mononuclear Cells (PBMCs) from Mice

Circulating total PBMCs were isolated by density-gradient centrifugation method using the Ficoll–Paque technique according to the manufacturer’s protocol. Briefly, 40 µL of 2% sodium citrate (pH 7.2) was injected into the left ventricle of deeply anesthetized mice and left to circulate for 1 min, then 0.5–1 mL of whole blood was drawn via inferior vena cava puncture using a 21-gauge syringe containing 40 µL of 4% sodium citrate. Within 2 h, blood samples were diluted into 1.5 mL using sterile PBS and laid carefully on top of 1.125 mL (a ratio of 3:4) of Ficoll-Paque-Premium 1.084 (Product code:17–5446-02, GE Healthcare life sciences) and centrifuged at 300× *g* for 30 min at room temperature. The PBMC layer was carefully aspirated and isolated PBMCs were washed twice in Gibco^®^ RPMI 1640 medium (Invitrogen-Thermo Fisher Scientific, Grand Island, NY, USA) and then stored in freezing medium (90% endotoxin-free FBS albumin in DMSO) at a density of 0.5 million cells/mL.

### 4.5. Western Blot Analysis

Retinas or retinal EC were lysed in RIPA buffer (Millipore, Billerica, MA, USA) and 30 µg of total protein was separated by SDS-PAGE. Antibodies used are listed in Table 2. Membranes were visualized using Pierce™ ECL Western Blotting Substrate system (Thermo Fisher Scientific) or Odyssey^®^ CLx Imaging System (LI-COR Biosciences) after incubation with their respective secondary antibodies. Band intensities were quantified using Alpa Innotech Fluorchem (Santa Clara, CA, USA) or image-J imaging and densitometry software and expressed as relative optical densitometry (ROD) compared to the WT-ND control group.

### 4.6. Human Retinal Endothelial Cell (REC) Culture Studies

Cells and supplies were purchased from Cell Systems Corporations (Kirkland, WA, USA) and VEC Technology (Rensselaer, NY, USA) as described previously [16]. Confluent cells were switched to serum-free medium for 6 h then treated for 12 h with palmitate coupled to BSA (Pal-BSA) solutions 400 μmol/L of Pal-BSA. Equal volumes of 50% ethyl alcohol solution without any palmitate dissolved in BSA served as a control.

### 4.7. Overexpression or Silencing of TXNIP Expression in Human REC

Overexpression or silencing of TXNIP expression using GFP-Homo sapiens-TXNIP plasmid (Origene, Rockville, MD) or TXNIP small interfering RNA (siRNA), respectively, was performed as described in [45] and [16], using Amaxa-nucleofector and primary EC kit following the manufacturer’s protocol (Lonza, Germany). For additional set of experiments, EC were treated with interleukin-1 receptor antagonist (IL-1RA; 100 ng/mL, R&D Systems) for 24 h.

### 4.8. Ex-Vivo Co-Culture Studies of REC and PBMC Determination of Leukostasis and EC-Death

Mouse REC cultures were grown in control medium (DMEM with 5 mM glucose) containing 10% serum as described by Sheibani’s group [46]. Isolated PBMCs were labeled with a fluorescent cell tracker (Cell-Tracker CM-DiI, Life technologies; 5 µg/mL in RPMI culture medium) and co-cultured with mouse retinal EC. Cells were incubated for either 2 h for assessment of leukostasis or 24 h for assessment of EC death using a terminal deoxynucleotidyl transferase dUTP nick end labeling assay kit (ApopTag®, Catalogue # S7160, Millipore) following the manufacturer’s protocol. The number of adherent PBMCs and TUNEL positive nuclei were counted from at least 3–4-different fields from each cultured well and calculated as the average number/field using AxioObserver.Z1 Microscope (Zeiss, Germany).

### 4.9. Determination of Retinal (Occluded) Acellular Capillaries

Retinal vasculature was isolated as described previously [39]. Retinal vasculature was stained with periodic acid-Schiff and hematoxylin. Acellular capillaries were identified as capillary-sized blood vessel tubes having no nuclei anywhere along their length. The number of acellular capillaries were counted from at least 3–6-different fields of the mid-retinal area and calculated as the average number/field using AxioObserver.Z1 Microscope (Zeiss, Dublin, CA, USA).

### 4.10. Vascular Morphology Analyses

Mice were systemically perfused with (FITC)-labeled ConA lectin. Retinal flat mounts were imaged using a series of Z-stack images (5–6 slices, 18–20 µm thickness) in at least 4–6 different-fields in the mid-retinal area spanning all three layers of retinal vasculature (innermost, intermediate and deep capillary plexus) using AxioObserver.Z1 Microscope (Zeiss, Germany). Z-stack images were compressed, transformed into one binary skeletonized picture and then relative branching density was calculated using FIJI software (NIH open access software) and expressed compared to WT-ND control group.

### 4.11. Statistical Analysis

Results were expressed as mean ± SEM. Statistical analyses was performed using Number Cruncher Statistical Software (NCSS, https://www.ncss.com/download/ncss) using three-way analysis of variance (a 2X2X2 ANOVA) followed by Holm–Sidak’s multiple comparison test for testing for differences among all experimental groups and for interaction between the type of diet (HFD versus normal diet), genotype (WT versus TKO mice) and gender (males versus females) for all in vivo experiments. Of note, initial three-way analyses indicated no differences in response or interaction between males and females for all experiments performed. Hence, data from both males and females within each experimental group were pooled together and a two-way ANOVA was performed using Graphpad-prizm-7 software for final analyses. Same statistical analyses were also applied for all other animal experiments. A two-way ANOVA was also used to test the interaction between the type of diet and genotype in vivo and the presence and absence of Pal-BSA across silencing TXNIP expression and between IL-1RA treatment and TXNIP overexpression in vitro. Significance was defined as *p* < 0.05.

## 5. Conclusions

Several previous studies had established a prominent role for TXNIP in mediating retinal NLRP3-inflammasome activation in other models of type-1 diabetes, hyperglycemia and retinal neurotoxicity. However, this is the first evidence that TXNIP is directly required for NLRP3-inflammasome activation and retinal microvascular dysfunction in a model of HFD-induced insulin resistance and pre-diabetes. Further studies are warranted to dissect the role of hyperglycemia versus saturated fatty acid and altered insulin signaling in diabetic versus obese pre-diabetic or insulin-resistant conditions.

## Figures and Tables

**Figure 1 ijms-21-03983-f001:**
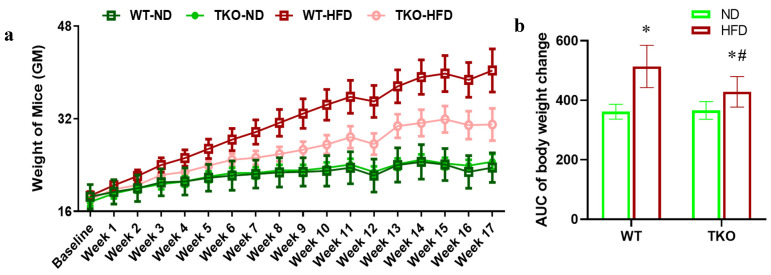
(**a**) High fat diet (HFD) significantly increased weight in both wild type (WT) and TXNIP knockout (TKO) mice. Six-week-old age and gender matched C57Bl/6J WT and TKO mice were randomized for feeding with either standard chow (normal diet; WT-ND and TKO-ND groups) or high fat diet (WT-HFD and TKO-HFD groups) for 18 weeks. Mice were weighed weekly and clearly HFD resulted in similar increases in weight gain in both WT-HFD and TKO-HFD mice when compared to their ND-controls over the 18 weeks (* *p* < 0.05, *n* = 11). (**b**) Area under the curve (AUC) analysis indicated that WT-HFD gained overall more weight when compared to TKO-HFD (# *p* < 0.05, *n* = 11).

**Figure 2 ijms-21-03983-f002:**
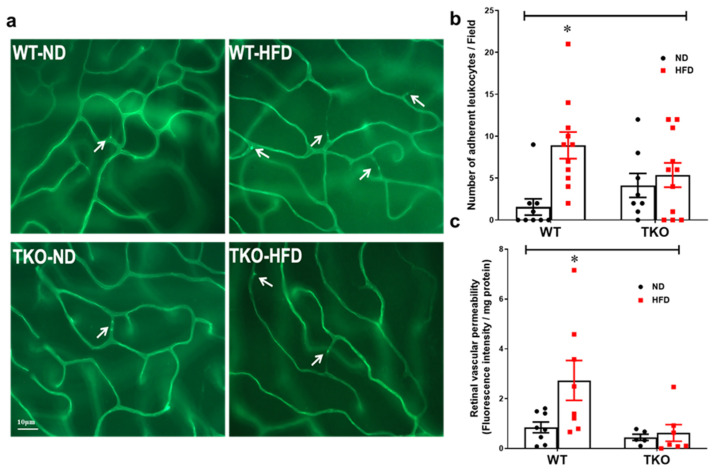
Deletion of TXNIP mitigates HFD-induced retinal leukostasis and BRB breakdown. (**a**) Representative pictures and (**b**) quantification of the number of adherent leukocytes occluding the poorly perfused retinal micro-vessels per field (indicated by white arrows) showed higher numbers in the WT-HFD group but had no significant effect on the TKO-HFD group compared with WT-ND group (*n* = 8–11 mice/group; * *p* < 0.05 vs. other groups). (**c**) Quantification of BSA-Fluorescence in the retina tissue was higher in the WT-HFD group whereas both TKO-ND and TKO-HFD groups had significantly lower levels of BSA-Fluorescence extravasation, respectively, compared with WT-ND group (*n* = 5–9 mice/group; * *p* < 0.05 vs. other groups).

**Figure 3 ijms-21-03983-f003:**
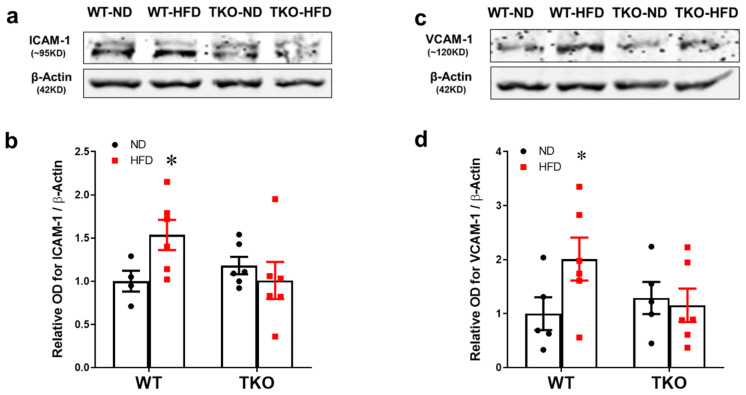
Deletion of TXNIP mitigates HFD-induced retinal expression of adhesion molecules. (**a**,**c**) Representative WB blots and statistical analyses of (**b**) ICAM-1 and (**d**) VCAM-1 showed increased expression of ICAM-1 and VCAM-1 levels in WT-HFD group compared with WT-ND group. In contrast, TKO mice groups showed no changes in response to HFD. Two-way ANOVA showed significant interaction between the type of diet and genotype across both ICAM-1 and VCAM-1 expression (*n* = 4–6 mice/group; * *p* < 0.05 vs. other groups).

**Figure 4 ijms-21-03983-f004:**
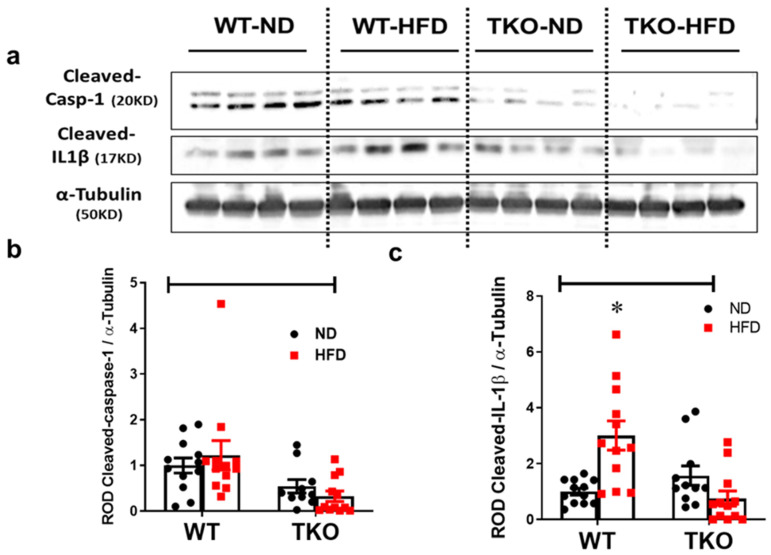
Deletion of TXNIP abrogates HFD-induced NLRP3-inflammasome activation in the retina. Representative WB blots (**a**) and statistical analyses of protein expression of retinal (**b**) cleaved caspase-1 and (**c**) cleaved IL-1β showed a significant increase in cleaved IL-1β expression, and a trend towards higher levels of cleaved caspase-1 in WT-HFD group compared with WT-ND. In contrast, TKO-ND and TKO-HFD groups showed either no changes or significantly lower levels of all target protein expression compared with WT-ND group, respectively. Two-way ANOVA showed significant interaction between the type of diet across cleaved IL-1β expression and between the genotype across expression of cleaved IL-1β and cleaved caspase-1 (*n* = 12 mice/group; * *p* < 0.05 vs. other groups).

**Figure 5 ijms-21-03983-f005:**
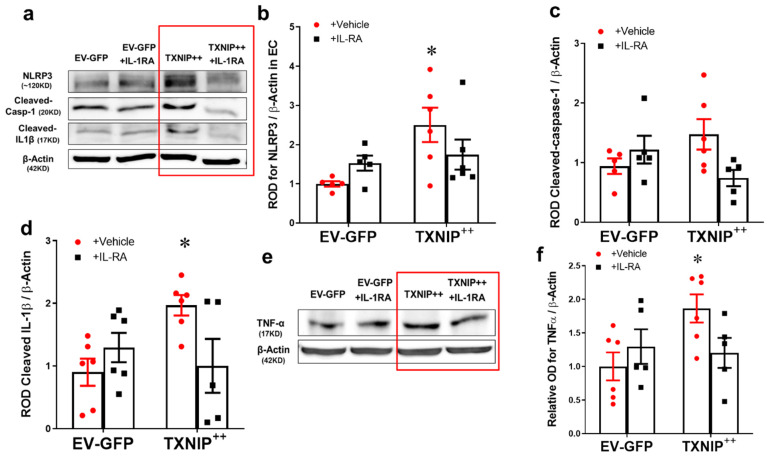
Overexpression of TXNIP activates NLRP3-inflammasome and EC inflammation in an IL-1β-dependent fashion. Representative WB blots (**a**,**e**) and statistical analysis of protein expression detected from REC lysates, of endothelial (**b**) NLRP3, (**c**) cleaved caspase-1, (**d**) cleaved IL-1β, and (**f**) TNF-α, showing significant increases in NLRP3, cleaved IL-1β and TNF-α and a trend towards increased expression of cleaved caspase-1 in the TXNIP overexpressing (TXNIP ^++^ + vehicle) group compared with the EV-GFP + vehicle control group. Treatment of TXNIP-overexpressing REC with 100 ng/mL of IL-1RA (TXNIP ^++^ + IL-1RA) suppressed the effect of TXNIP overexpression on inflammasome activation and TNF-α expression. Two-way ANOVA showed significant interaction between IL-1RA and TXNIP overexpression across NLRP3 and cleaved IL-1β expression levels (*n* = 5–6, each measurement was performed in triplicate using two different cell preparations; * *p* < 0.05 vs. EV-GFP + vehicle group).

**Figure 6 ijms-21-03983-f006:**
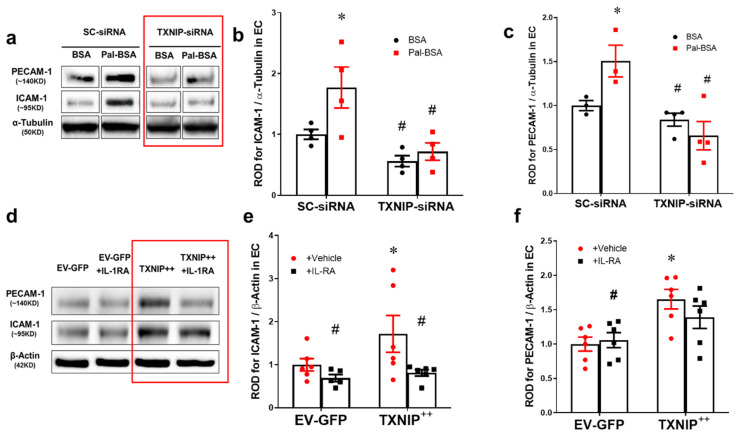
Silencing of TXNIP abolishes, while its overexpression induces, REC adhesion molecule upregulation in an IL-1β-dependent fashion. (**a**) Representative WB blots (boxes denote separate lanes from the same gel) and statistical analyses of (**b**) ICAM-1 and (**c**) PECAM-1 protein expression showed higher levels in cells incubated with PAL-BSA compared with BSA controls in the scrambled-siRNA group. Silencing TXNIP expression using siRNA abolished such responses. Two-way ANOVA showed significant interaction between Pal-BSA and silencing TXNIP expression across PECAM-1 levels (*n* = 3–4, each measurement was performed in duplicate using two different cell preparations; * *p* < 0.05 vs. SCsiRNA + BSA, # vs. SCsiRNA + Pal-BSA). (**d**) Representative blots and WB analysis of (**e**) ICAM-1 and (**f**) PECAM-1 protein expression showed higher levels in TXNIP overexpression (TXNIP ^++^ + vehicle) group compared with EV-GFP + vehicle control group, respectively. Treatment of TXNIP-overexpressing REC with 100 ng/mL of IL-1RA (TXNIP ^++^ + IL-1RA) suppressed the effect of TXNIP overexpression on expression of REC adhesion molecules (*n* = 5–6, each measurement was performed in triplicate using two different cell preparations; * *p* < 0.05 vs. EV-GFP+ vehicle, # vs. TXNIP ^+ +^ + vehicle).

**Figure 7 ijms-21-03983-f007:**
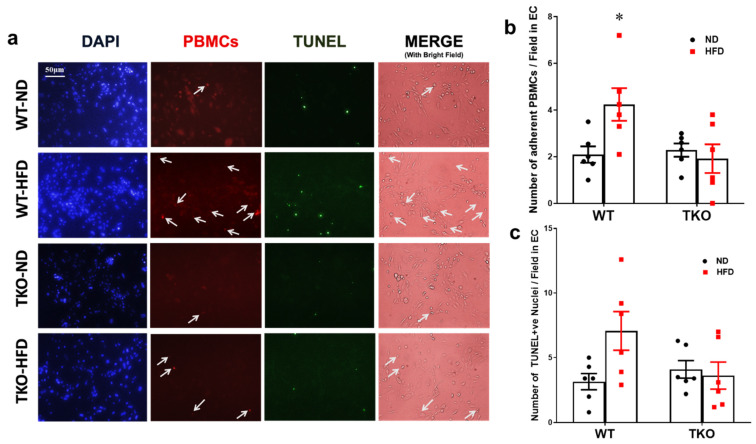
Deletion of TXNIP in leukocytes prevents leukostasis and REC death. (**a**) Representative images and (**b**) quantification of the number of adherent PBMCs (indicated by white arrows) to cultured mouse REC per field after 2 hours and (**c**) the number of TUNEL-positive nuclei per field after 24 h; showed significantly higher numbers of adherent PBMCs and a trend in increased numbers of TUNEL-positive nuclei in the WT-HFD group compared with WT-ND control group, respectively. In contrast, both TKO-ND and TKO-HFD groups showed no significant changes compared with WT-ND control group. Two-way ANOVA showed a significant interaction between the genotype across the number of adherent PBMCs and between the type of diet across the number of adherent PBMCs and the number of TUNEL-positive nuclei (*n* = 5–6, each measurement was performed in triplicate using two different cell preparations; * *p* < 0.05 vs. other groups).

**Figure 8 ijms-21-03983-f008:**
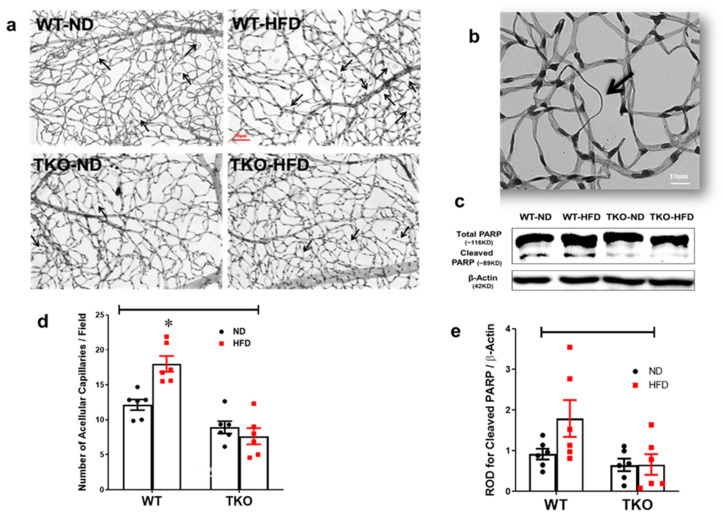
Deletion of TXNIP abrogates HFD-induced retinal microvascular degeneration. (**a**) Representative pictures of degenerated acellular capillaries from each of the groups, (**b**) magnification of a degenerated acellular capillary showing loss of endothelial cell nucleus and the remaining thickened basement membrane and (**c**) quantification of the number of acellular capillaries per field showed significantly higher numbers in the WT-HFD group compared with WT-ND. Two-way ANOVA showed significant interaction between the type of diet and genotype across microvascular degeneration (*n* = 6 mice/group; * *p* < 0.05 vs. all groups). (**d**) Representative WB blots and (**e**) statistical analyses of cleaved PARP showed a trend towards increased expression in the WT-HFD group compared with the WT-ND control group. Two-way ANOVA showed significant interaction between genotype (TKO vs. WT) (*n* = 8 mice/group).

**Figure 9 ijms-21-03983-f009:**
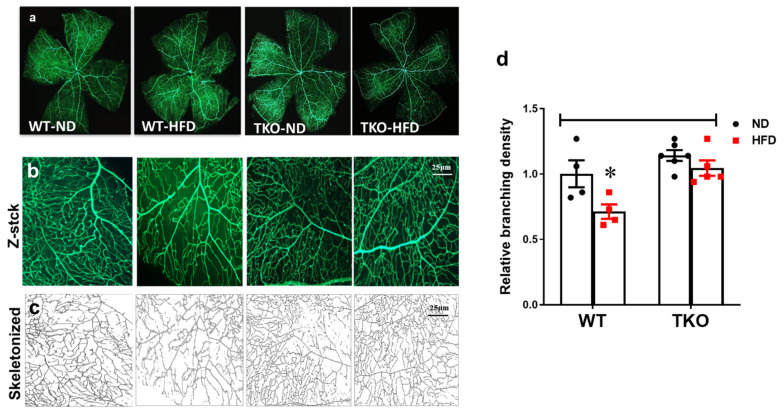
Deletion of TXNIP abrogates HFD-induced microvascular morphological changes. Representative images of (**a**) total retinal flat mounts stitched from individual lopes of the retina, (**b**) representative of a compressed Z-stack image of one of the retina loops and (**c**) skeletonized compressed images from each of the animal groups. (**d**) WT-HFD mice had decreased retinal vascular branching density compared with WT-ND control group. In contrast, TKO mice had a comparable morphological appearance compared to the WT-ND group that did not change in response to HFD. Two-way ANOVA showed significant interaction between the type of diet and genotype across microvascular density (*n* = 4–6 mice/group; * *p* < 0.05 vs. other groups).

**Table 1 ijms-21-03983-t001:** Summary of the fasting blood glucose (FBG) levels of all animal groups. (Data are represented as mean ± SEM, *n* = 11–19 / group; * *p*-value < 0.05 vs. WT-ND, and ¶ vs. WT-HFD), ≠ vs. TKO-ND.

Animal Group	FBG (mg/dl)		
	8 Weeks	12 Weeks	18 Weeks
**WT-ND**	128.8 (± 7.95)	126.91 (± 3.14)	118.27 (± 5.07)
**WT-HFD**	154.12 * (± 6.61)	130.71 (± 2.85)	135.29 * (± 5.84)
**TKO-ND**	85.98 *^¶^ (± 7.51)	70.31 *^¶^ (± 5.49)	69.88 *^¶^ (± 4.74)
**TKO-HFD**	96.59 *^¶^ (± 8.55)	82.00 *^¶^ (± 3.32)	91.81 *^¶≠^ (± 3.93)

**Table 2 ijms-21-03983-t002:** List of antibodies and sources used to detect protein expression by Western blot. All antibodies were obtained commercially and validated by its manufacturer.

Antibody	Source	Catalogue #	Company
**TXNIP**	Polyclonal	403700	Invitrogen-Thermo-Fischer Scientific, Waltham, MA
**NLRP-3**	Polyclonal	LS-B4321	LifeSpan Biosciences, Inc., Seatle, WA
**IL1β**	Polyclonal	ab9722	Abcam, Cambridge, MA
**TNF-a**	Polyclonal	ab9635	Abcam, Cambridge, MA
**Tubulin**	Monoclonal	ab4074	Abcam, Cambridge, MA
**GAPDH**	Polyclonal	5174	Cell Signaling Tech, Danvers, MA
**Caspase-1**	Monoclonal	24232	Cell Signaling Tech, Danvers, MA
**ICAM-1**	Monoclonal	67836	Cell Signaling Tech, Danvers, MA
**PECAM-1**	Monoclonal	77699	Cell Signaling Tech, Danvers, MA
**PARP**	Polyclonal	9542	Cell Signaling Tech, Danvers, MA
**VCAM-1**	Monoclonal	sc-13160	Santa Cruz Biotech, Dallas, TX

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
