# Peer review of "Deletion of Thioredoxin-Interacting Protein (TXNIP) Abrogates High Fat Diet-Induced Retinal Leukostasis, Barrier Dysfunction and Microvascular Degeneration in a Mouse Obesity Model"

_ijms, 2020, doi:10.3390/ijms21113983_

Round 1

Reviewer 1 Report

This article try to clarify the role of thioredoxin interacting protein (TXNIP) involved in retinal early microvascular dysfunction. It is an advanced and novelty topic, however, the evidence in the manuscript is not strong enough and the quality of data needs to improve. The following are my suggestions as below.

Major:

  1. The resolution of Figures is poor. It is difficult to read the words inside Figures clearly. All immunohistochemical photographs have no scale bar.
  2. Although statistical analyses of retinal cleaved-caspase-1 have no difference between WT and TKO with normal diet (Fig. 4b), the result of WB photograph seems to show the bands of cleaved-caspase-1 in the WT-ND have more increase than those in the TK-ND (Fig. 4a). Does it have possibility that the levels of cleaved IL-1β in the WT are more than those in the TKO under normal physical condition? By the way, the authors should mention when this test had been operated after mice fed with HFD (after 8-weeks?) and there is no description about symbol # (Fig. 4c).
  3. There are no marked arrows to indicate leukostasis in the “Merge” lane (Fig. 7a). Please explain how to calculate data. By the way, why is the background color of “Merge” pink?
  4. The data of Fig. 8a do not indicate which group it was and include all of groups. In order to confirm the over-expression of TXNIP induced by HFD in WT mice, immunohistochemical stain and WB data should be considered to operate at early stage of 8-weeks or end-point of 18-weeks. There is an obvious displacement mistake in Fig. 8 figure legend (a) Representative WB blots and (b) statistical analyses of cleaved PARP and (c) Representative picture and (d) quantification of the number.

Minor:

  1. The number of each animal group is unclear and is difficult to follow. For example, fasting blood glucose (n=11-19) do not match body weight (n=11) or leukostasis (Fig. 7a-b, P<0.05, n=6) and retinal microvascular degeneration (Fig. 8a-b, P<0.05, n=6). The authors should mention how to arrange use of animals in the “Materials and Methods”.
  2. The authors mention “The Deletion of TXNIP significantly prevented leukostasis in TKO-HFD compared with WT-ND and TKO-ND groups.” (line 92-93), but there is no p value within content and marks within Fig. 2.
  3. There are some typos need to correct them. Such as tumor necrosis factor-alpha (TNF- ) (line 38), Ex-vivo (line 67), IL (line 122) (line 300), (P<0.5, n=4-6) (line 142), cleaved IL-1b and TNF-a (line 156), (P<0.5, n=4) (line 169), # vs S SCsiRNA+Pal-BSA (line 183)
  4. 26 has been published as below:

Elshaer, S.L.; Mohamed, I.N.; Coucha, M.; Altantawi, S.; Eldahshan, W.; Bartasi, M.L.; Shanab, A.Y.; Lorys, R.; El-Remessy, A.B. Deletion of txnip mitigates high-fat diet-impaired angiogenesis and prevents inflammation in a mouse model of critical limb ischemia. Antioxidants 2017, 6.

Author Response

Reviewer 1

We would like to thank the reviewer for taking the time to review our manuscript, describing it as “an advanced and novelty topic” and providing helpful suggestions that were instrumental in improving the quality of the manuscript. We have addressed the criticism raised by the reviewer as detailed below.

Major:

  • The resolution of Figures is poor. It is difficult to read the words inside Figures clearly. All immunohistochemical photographs have no scale bar.

We do apologize as the first submission was in a PDF format with lower resolution. Revised and better-quality images with the scale bar are now submitted.

  • Although statistical analyses of retinal cleaved-caspase-1 have no difference between WT and TKO with normal diet (Fig. 4b), the result of WB photograph seems to show the bands of cleaved-caspase-1 in the WT-ND have more increase than those in the TK-ND (Fig. 4a). Does it have possibility that the levels of cleaved IL-1β in the WT are more than those in the TKO under normal physical condition?

Although the representative figures show a slight trend towards decreased IL-1b between WT-ND & TKO-ND groups, however, as shown in the bar graph, the overall averages of all samples from all groups showed no statistically significant differences between WT & TKO mice under normal conditions. In addition, our prior published work that involved detection of IL-1b in WT and TKO showed no significant difference related to TXNIP deletion in HFD model ((Mohamed, Sarhan et al. 2018), (Elshaer, Mohamed et al. 2017)), neurotoxic model (El-Azab, Baldowski et al. 2014), or  ischemia reperfusion model (Coucha, Mohamed et al. 2017).

  • By the way, the authors should mention when this test had been operated after mice fed with HFD (after 8-weeks?) and there is no description about symbol # (Fig. 4c).

We apologize for this mistake. Results pertained to Fig. 4 are observed at 8weeks of HFD feeding. Also, the # has been removed from the figure. We have clarified the time point for each results section & corresponding figures. Please see updated text of the revised manuscript.

  • There are no marked arrows to indicate leukostasis in the “Merge” lane (Fig. 7a).

We thank the reviewer for his comment. Please see updated Fig. 7a with marked arrows.

  1. Please explain how to calculate data.

The number of adherent PBMCs and TUNEL positive nuclei were counted from at least 3-4-different fields from each cultured well and calculated as the average number/field using AxioObserver.Z1 Microscope (Zeiss, Germany). This clarification is now added to method section.

  1. By the way, why is the background color of “Merge” pink?

We thank the reviewer for his comment. The pink color is due to the Merge of all channels with the bright field channel which helped to confirm the morphology and the outline of the retinal endothelial cells.

  • The data of Fig. 8a do not indicate which group it was and include all of groups.

We thank the reviewer for his comment. A new representative of all the 4 animal groups is now added in revised Fig. 8a.

  • In order to confirm the over-expression of TXNIP induced by HFD in WT mice, immunohistochemical stain and WB data should be considered to operate at early stage of 8-weeks or end-point of 18-weeks.

Our previous studies have confirmed that HFD triggers the upregulation of retinal TXNIP expression and direct interaction the NLRP3 inflammasome in a rat model of HFD, that was co-localized within the retinal microvasculature (Mohamed, Hafez et al. 2014). In the current study, western blot analysis has further confirmed the HFD-induced upregulation of TXNIP at 8-weeks of HFD. Please see updated Western blot data of TXNIP at 8-weeks of HFD is shown in supplementary figure 1.

  • There is an obvious displacement mistake in Fig. 8 figure legend (a) Representative WB blots and (b) statistical analyses of cleaved PARP and (c) Representative picture and (d) quantification of the number.

We do apologize for the oversight. Please see corrected Fig. 8 legend in the revised manuscript.

Minor:

  • The number of each animal group is unclear and is difficult to follow. For example, fasting blood glucose (n=11-19) do not match body weight (n=11) or leukostasis (Fig. 7a-b, P<0.05, n=6) and retinal microvascular degeneration (Fig. 8a-b, P<0.05, n=6). The authors should mention how to arrange use of animals in the “Materials and Methods”.

As described in the method section, all WT and TKO animals were produced using in-house breading in our facility. Therefore, different littermate batches from all WT and TKO animals were randomized for different experiments whenever they became available. We have also made sure to include animals from different litter-groups in all performed experiments. This way, we were able to maintain a balance between using sibling pairs and reduce any unforeseen bias that might be associated with one litter versus another (i.e: any potential inter-litter variations). This clarification is added to the method section.

  • The authors mention “The Deletion of TXNIP significantly prevented leukostasis in TKO-HFD compared with WT-ND and TKO-ND groups.” (line 92-93), but there is no p value within content and marks within Fig. 2.

We do apologize for the oversight. It is now corrected.

  • There are some typos need to correct them. Such as tumor necrosis factor-alpha (TNF- ) (line 38), Ex-vivo (line 67), IL (line 122) (line 300), (P<0.5, n=4-6) (line 142), cleaved IL-1b and TNF-a (line 156), (P<0.5, n=4) (line 169), # vs S SCsiRNA+Pal-BSA (line 183)

We do apologize for the oversight. These are now corrected.

  • Ref 26 has been published as below: Elshaer, S.L…

We do apologize for the oversight. It is now corrected.

Reviewer 2 Report

In this study, the authors compared the responses of wild type C57 mice and TXNIP knockout mice, both at 6 weeks of age, to high fat diet (HFD) for up to 18 weeks. A striking finding (Fig. 9) is that wild type mice had 30% reduction in retinal vascular branching density. Knockout TXNIP prevented retinal microvascular degeneration. In addition, the authors performed a number of in vivo and in vitro mechanism studies. Based on the available data they proposed that HFD-induced upregulation of adhesion molecules, leukostasis and inflammasome activation contributed to the retinal vascular lesions, and these pathological events were effectively suppressed after TXNIP deletion. The HFD model is a commonly used model; the findings from the current study can have broad impact to obesity and diabetes research.

Despite the potential impact, the current draft of the manuscript is missing data from one key experiment. If leukostasis is one of the most important mechanisms, data on the infiltration of inflammatory cells (or lack of infiltration) in wild type and TXINP knockout retina will have to be presented in greater detail than the preliminary findings of Fig. 2a. Additional comments on experimental design and data interpretation are listed below.

  1. Line 38, (TNF-) “alpha” was not displayed
  2. Reference 26 is published. Please update the citation
  3. Table 1, TXINP knockout mice on normal diet had much lower fasting blood glucose. Did the authors measure plasma triglyceride, as reported in Ref 28?
  4. Fig 2b and 2c, the comparisons appeared to be between wild type control and TXINP knockout on HFD. If this is not what the authors intended to show, please correct
  5. Fig 2c, the authors made an important conclusion that deletion of TXINP protected the integrity of the blood-retinal barrier (BRB). During the experiments, however, retina tissues were collected 20 min after the injection of fluorescein-conjugated BSA. The methodology will not distinguish whether the BSA was inside or outside the retinal blood vessels. An independent validation assay, such as in vivo fluorescein angiography, or Evans dye permeation assay with much longer circulation time, will be needed in order to further strengthen the conclusion
  6. Fig 4, western blots showed cleaved IL-1 beta and cleaved caspase 1. It will be highly informative if the authors can present data on the non-cleaved forms of caspase 1 and IL-1 beta. Based on the data presented, it is unclear whether TXINP knockout downregulated the expression of these proteins in the retina
  7. Fig. 5a, control experiment should be performed to check the expression levels of TXINP and GFP after IL-1R antagonist treatment
  8. Fig. 5, in cell culture experiments, did the authors measure TNF-alpha and IL-1 beta that had been released into the conditioned medium? It was not specified in either the text or the figure legend
  9. Fig. 6, control experiment will be needed to demonstrate the endogenous level of TXNIP before and after siRNA transfection
  10. Fig. 7, the assay design can be improved if the retinal endothelial cells are pre-labeled with a different fluorescent dye before PBMNCs are added

Author Response

Reviewer 2

We would like to thank the reviewer for taking the time to review our manuscript, describing it as “striking finding and it can have broad impact to obesity and diabetes research” and providing helpful suggestions that were instrumental in improving the quality of the manuscript. Please find our responses below.

Despite the potential impact, the current draft of the manuscript is missing data from one key experiment. If leukostasis is one of the most important mechanisms, data on the infiltration of inflammatory cells in wild type and TXINP knockout retina will have to be presented in greater detail than the preliminary findings of Fig. 2a.

The data presented using leukostasis assay is a quantification of the number of infiltered leukocytes adherent to retinal endothelium. This is a sophisticated and highly laborious technique that is widely utilized to determine leukostasis by our group (Al-Shabrawey, Rojas et al. 2008) and others ((Thounaojam, Montemari et al. 2019), (Bretz, Savage et al. 2015), (Noda, Nakao et al. 2014)). In addition, the results from the leukostasis are complimented with determining the expression of adhesion molecules (ICAM-1, PECAM-1, and VECAM-1) and expression of inflammasome and inflammatory mediators both in vivo and in vitro that are fundamental for the recruitment & adhesion of leukocytes on the endothelial cell surface.

Additional comments on experimental design and data interpretation are listed below.

  • Line 38, (TNF-) “alpha” was not displayed.

We do apologize for the oversight. It is now corrected.

  • Reference 26 is published. Please update the citation.

We do apologize for the oversight. It is now corrected.

  • Table 1, TXINP knockout mice on normal diet had much lower fasting blood glucose. Did the authors measure plasma triglyceride, as reported in Ref 28?

We would like to thank the reviewer for his comment. This is part of the comprehensive phenotype characterization of the TXNIP-KO mice was published by the founder of the TXNIP-KO mice and other groups ((Hui, Andres et al. 2008) and (Yoshihara, Fujimoto et al. 2010)). Furthermore, our group have also characterized the metabolic profile of TKO (controls and HFD) at 8-weeks of HFD that was previously published (Elshaer, Mohamed et al. 2017).

  • Fig 2b and 2c, the comparisons appeared to be between wild type control and TXINP knockout on HFD. If this is not what the authors intended to show, please correct.

We do apologize for the oversight. Fig 2. Legend is now corrected to clearly describe that comparison is between wild-type ND and wild-type HFD rather than the comparison between WT and TKO.

  • Fig 2c, the authors made an important conclusion that deletion of TXINP protected the integrity of the blood-retinal barrier. During the experiments, however, retina tissues were collected 20 min after the injection of fluorescein-conjugated BSA. The methodology will not distinguish whether the BSA was inside or outside the retinal blood vessels. An independent validation assay, such as in vivo fluorescein angiography, or Evans dye permeation assay will be needed to further strengthen the conclusion.

We would like to thank the reviewer for his comment. As described in the method section, in order to compensate for that, the retinal fluorescence was normalized to that detected in both the serum and to the total protein content of each sample. In addition, this limitation of the assay described is equally distributed between all groups in both HFD (disease) and ND (control), hence, this should not impact the statistical comparison. Furthermore, we would like to point out that this technique is a standard, well-accepted technique for BRB detection used by our group & others ((Elshaer, Alwhaibi et al. 2019), (Mohamed, Coucha et al. 2018), (Mysona, Al-Gayyar et al. 2013), (Al-Shabrawey, Rojas et al. 2008) & (Antonetti, Barber et al. 1998)), and is  equally rigorous  to  the Evans blue or angiography assays.

  • Fig 4, western blots showed cleaved IL-1 beta and cleaved caspase 1. It will be highly informative if the authors can present data on the non-cleaved forms of caspase 1 and IL-1 beta. Based on the data presented, it is unclear whether TXINP knockout downregulated the expression of these proteins in the retina.

We would like to thank the reviewer for his comment. Our prior published work that involved detection of IL-1b in WT and TKO showed no significant difference related to TXNIP deletion in HFD model ((Mohamed, Sarhan et al. 2018), (Elshaer, Mohamed et al. 2017)), neurotoxic model (El-Azab, Baldowski et al. 2014), or  ischemia reperfusion model (Coucha, Mohamed et al. 2017).

  • 5a, control experiment should be performed to check the expression levels of TXINP and GFP after IL-1R antagonist treatment.

We would like to thank the reviewer for his suggestion. A representative of GFP transfection efficacy and TXNIP expression are now shown in supplementary Figure-3. As shown in the figure, these results showed ~ 2.5 fold increase in TXNIP expression from cells transduced with TXNIP plasmid (TXNIP++ ) compared to EV-GFP controls. Interestingly, treatment with IL-1R antagonist tended to reduce TXNIP expression but didn’t reach statistical significance in both EV-GFP control and TXNIP++.

  • 5, in cell culture experiments, did the authors measure TNF-alpha and IL-1 beta that had been released into the conditioned medium? It was not specified in either the text or the figure legend.

We do apologize for the oversight. Please see revised manuscript where it is clarified that it was measured in cell lysate in the corresponding results and Fig. Legend sections.

  • 6, control experiment will be needed to demonstrate the endogenous level of TXNIP before and after siRNA transfection.

We would like to thank the reviewer for his comments and to explain that this is a routine procedure in the lab and we have previously published the efficacy of TXNIP-siRNA transfection in in (Mohamed, Hafez et al. 2014) and (Abdelsaid, Matragoon et al. 2013).

  • 7, the assay design can be improved if the retinal endothelial cells are pre-labeled with a different fluorescent dye before PBMNCs are added.

We would like to thank the reviewer for his insightful suggestion. The PMNCs were freshly isolated from various animal groups and we apologize that we do not currently have animals on HFD study to repeat this experiment. However, the bright field channel was used to confirm the morphology and the outline of the EC, which is shown in the “Merge” column of all channels. Please see updated Fig. 7.

Cited references

Abdelsaid, M. A., S. Matragoon and A. B. El-Remessy (2013). "Thioredoxin-interacting protein expression is required for VEGF-mediated angiogenic signal in endothelial cells." Antioxid Redox Signal 19(18): 2199-2212.

Al-Shabrawey, M., M. Rojas, T. Sanders, A. Behzadian, A. El-Remessy, M. Bartoli, A. K. Parpia, G. Liou and R. B. Caldwell (2008). "Role of NADPH oxidase in retinal vascular inflammation." Invest Ophthalmol Vis Sci 49(7): 3239-3244.

Antonetti, D. A., A. J. Barber, S. Khin, E. Lieth, J. M. Tarbell and T. W. Gardner (1998). "Vascular permeability in experimental diabetes is associated with reduced endothelial occludin content: vascular endothelial growth factor decreases occludin in retinal endothelial cells. Penn State Retina Research Group." Diabetes 47(12): 1953-1959.

Bretz, C. A., S. R. Savage, M. E. Capozzi, S. Suarez and J. S. Penn (2015). "NFAT isoforms play distinct roles in TNFalpha-induced retinal leukostasis." Sci Rep 5: 14963.

Coucha, M., I. N. Mohamed, S. L. Elshaer, O. Mbata, M. L. Bartasis and A. B. El-Remessy (2017). "High fat diet dysregulates microRNA-17-5p and triggers retinal inflammation: Role of endoplasmic-reticulum-stress." World J Diabetes 8(2): 56-65.

El-Azab, M. F., B. R. Baldowski, B. A. Mysona, A. Y. Shanab, I. N. Mohamed, M. A. Abdelsaid, S. Matragoon, K. E. Bollinger, A. Saul and A. B. El-Remessy (2014). "Deletion of thioredoxin-interacting protein preserves retinal neuronal function by preventing inflammation and vascular injury." Br J Pharmacol 171(5): 1299-1313.

Elshaer, S. L., A. Alwhaibi, R. Mohamed, T. Lemtalsi, M. Coucha, F. M. Longo and A. B. El-Remessy (2019). "Modulation of the p75 neurotrophin receptor using LM11A-31 prevents diabetes-induced retinal vascular permeability in mice via inhibition of inflammation and the RhoA kinase pathway." Diabetologia 62(8): 1488-1500.

Elshaer, S. L., I. N. Mohamed, M. Coucha, S. Altantawi, W. Eldahshan, M. L. Bartasi, A. Y. Shanab, R. Lorys and A. B. El-Remessy (2017). "Deletion of TXNIP Mitigates High-Fat Diet-Impaired Angiogenesis and Prevents Inflammation in a Mouse Model of Critical Limb Ischemia." Antioxidants (Basel) 6(3).

Hui, S. T., A. M. Andres, A. K. Miller, N. J. Spann, D. W. Potter, N. M. Post, A. Z. Chen, S. Sachithanantham, D. Y. Jung, J. K. Kim and R. A. Davis (2008). "Txnip balances metabolic and growth signaling via PTEN disulfide reduction." Proc Natl Acad Sci U S A 105(10): 3921-3926.

Mohamed, I. N., S. S. Hafez, A. Fairaq, A. Ergul, J. D. Imig and A. B. El-Remessy (2014). "Thioredoxin-interacting protein is required for endothelial NLRP3 inflammasome activation and cell death in a rat model of high-fat diet." Diabetologia 57(2): 413-423.

Mohamed, I. N., N. R. Sarhan, M. A. Eladl, A. B. El-Remessy and M. El-Sherbiny (2018). "Deletion of Thioredoxin-interacting protein ameliorates high fat diet-induced non-alcoholic steatohepatitis through modulation of Toll-like receptor 2-NLRP3-inflammasome axis: Histological and immunohistochemical study." Acta Histochem 120(3): 242-254.

Mohamed, R., M. Coucha, S. L. Elshaer, S. Artham, T. Lemtalsi and A. B. El-Remessy (2018). "Inducible overexpression of endothelial proNGF as a mouse model to study microvascular dysfunction." Biochim Biophys Acta Mol Basis Dis 1864(3): 746-757.

Mysona, B. A., M. M. Al-Gayyar, S. Matragoon, M. A. Abdelsaid, M. F. El-Azab, H. U. Saragovi and A. B. El-Remessy (2013). "Modulation of p75(NTR) prevents diabetes- and proNGF-induced retinal inflammation and blood-retina barrier breakdown in mice and rats." Diabetologia 56(10): 2329-2339.

Noda, K., S. Nakao, S. Zandi, D. Sun, K. C. Hayes and A. Hafezi-Moghadam (2014). "Retinopathy in a novel model of metabolic syndrome and type 2 diabetes: new insight on the inflammatory paradigm." FASEB J 28(5): 2038-2046.

Thounaojam, M. C., A. Montemari, F. L. Powell, P. Malla, D. R. Gutsaeva, A. Bachettoni, G. Ripandelli, A. Repossi, A. Tawfik, P. M. Martin, F. Facchiano and M. Bartoli (2019). "Monosodium Urate Contributes to Retinal Inflammation and Progression of Diabetic Retinopathy." Diabetes 68(5): 1014-1025.

Yoshihara, E., S. Fujimoto, N. Inagaki, K. Okawa, S. Masaki, J. Yodoi and H. Masutani (2010). "Disruption of TBP-2 ameliorates insulin sensitivity and secretion without affecting obesity." Nat Commun 1: 127.

Reviewer 3 Report

The authors have largely addressed the reviewers’ comments. However, two of the second reviewer’s criticisms still need what should be minor modifications.

1) Figure S3: The quality of the TXNIP is of low quality, which led the authors to present it with far too much contrast. As it is displayed, expression of TXNIP is the same in IL-1RA-treated control transfections and in TXNIP overexpressing cells (both shown conditions). However, the quantification indicates increased expression at least without IL-1RA. Better quality immunoblots should be shown that reflect the quantification.

2) The authors’ rebuttal letter contains a Supp Fig. 4. It is not clear to me whether it has been included in the manuscript or not as it has neither been referenced nor attached to the manuscript. However, the effectiveness of an siRNA treatment needs to be demonstrated. That’s really a minimal requirement for any siRNA experiment. A historical control is not sufficient. Such a panel should be added.

Author Response

  • Figure S3: The quality of the TXNIP is of low quality, which led the authors to present it with far too much contrast. As it is displayed, expression of TXNIP is the same in IL-1RA-treated control transfections and in TXNIP overexpressing cells (both shown conditions). However, the quantification indicates increased expression at least without IL-1RA. Better quality immunoblots should be shown that reflect the quantification.

A different representative is now shown in Fig.S3

We would like to provide a general clarification about WB, the mouse retina is a very tiny tissue that a maximum of 100ug protein is obtained. To be able to check the expression of multiple proteins, we had to probe the membranes several times with antibodies or cut some into strips to maximize the utilization of retina samples.  

2) The authors’ rebuttal letter contains a Supp Fig. 4. It is not clear to me whether it has been included in the manuscript or not as it has neither been referenced nor attached to the manuscript. However, the effectiveness of a siRNA treatment needs to be demonstrated. That’s really a minimal requirement for any siRNA experiment. A historical control is not sufficient. Such a panel should be added.

Supplementary Fig.4 showing the siRNA transfection efficiency data is under Appendix

Round 2

Reviewer 1 Report

No other specific comment for the revised version.

Author Response

No other specific comment for the revised version.

Thank you for your consideration.

Reviewer 2 Report

The authors have responded to some but not all of the previous comments and critiques. The new data (Supplementary Fig. 3) are not supportive to their conclusions. Some of the methodologies are still flawed. Unfortunately, the manuscript cannot be recommended for further consideration.

  • Fig. S3, cells transfected with EV-GFP and treated with IL-1RA showed marked increase in TXNIP, to a level similar to TXNIP overexpression. The data are contradictory to the conclusion of Fig. 5, that overexpression of TXNIP activated inflammasome in an IL-1 beta-dependent fashion. Increased TNXIP alone (in EV-GFP cells) was not sufficient to cause NLRP3 inflammasome activation. Furthermore, TXNIP+IL-1RA did not lower TXNIP (Fig. S3), but inhibited caspase 1 and IL-1 beta processing. Basically, the protein level of TXNIP had no association with inflammasome activation.
  • Fig. 6 does not show how siRNA transfection affected the protein (or RNA) level of the targeted protein.
  • Fig. 2C, measuring RBB permeability by fluorescein-conjugated BSA. Retina was harvested 20 min after perfusion. The time was too short to allow the diffusion of BSA into the tissue.
  • Fig. 2A, the leukostasis has served the purpose of the experiment, but does not show whether the inflammatory cells have (or have not) migrated into the retinal tissue where they can exert their functions.

Author Response

Fig. S3, cells transfected with EV-GFP and treated with IL-1RA showed marked increase in TXNIP, to a level similar to TXNIP overexpression. The data are contradictory to the conclusion of Fig. 5, that overexpression of TXNIP activated inflammasome in an IL-1 beta-dependent fashion. Increased TNXIP alone (in EV-GFP cells) was not sufficient to cause NLRP3 inflammasome activation. Furthermore, TXNIP+IL-1RA did not lower TXNIP (Fig. S3), but inhibited caspase 1 and IL-1 beta processing. Basically, the protein level of TXNIP had no association with inflammasome activation.

We respectfully disagree, looking at the data presented in Fig.3S/Fig.5, the findings support a positive association of increased TXNIP expression (TXNIP++), activation of inflammasome evident by increased expression of NLRP-3 and release of IL-1b (shown in Fig. 5). Inhibiting IL-1 Receptor signal showed a trend towards increased TXNIP expression in the EV-GFP cells, however, it was not statistically significant from neither the control EV-GFP cells. Only TXNIP overexpressing cells treated with Vehicle showed statistically significant levels of TXNIP expression that was attenuated upon IL-1RA treatment. More importantly, the same trend was also observed in parallel in the levels of all the target proteins studied, showing a very similar but also non-significant trend of increased levels of NLRP3, cleaved Caspase-1, Cleaved IL-1 beta and TNFa levels in the EV-GFP cells. Similarly, these targets were also significantly increased only in the TXNIP overexpressing cells treated with Vehicle and were also attenuated upon IL-1RA treatment. Perhaps, this might be possibly attributed to a possible basal partial agonist effect that might be exerted by IL-1RA on the cultured endothelial cells.

Fig. 6 does not show how siRNA transfection affected the protein (or RNA) level of the targeted protein.

Performing genetic inhibition studies using siRNA in general (Al-Gayyar et al 2011, Shanab et al 2015, and Shanab et al 2015) and for inhibiting TXNIP in cultured retinal endothelial cells is a routine procedure in our lab {Abdelsaid, 2013 #36} and {Mohamed, 2014 #83}. Please see additional data from actual experiment showing successful reduction of TXNIP levels using siRNA in cultured retinal endothelial cells.

Fig. 2C, measuring RBB permeability by fluorescein-conjugated BSA. Retina was harvested 20 min after perfusion. The time was too short to allow the diffusion of BSA into the tissue.

The fluorescein-conjugated BSA assay for assessing the retinal microvascular permeability is a standard, well-accepted technique that is inherently more sensitive compared to other classic techniques like Evans blue assays that requires 1 to 2 hours. The perfusion time is a function of the molecular weight of the tracer and the technique used to assess the level of the tracer. For example, sodium fluorescein is assessed within 1-10 minutes in rodent similar to human studies (PMID: 32352608, PMID: 22820291). In addition, previous studies by other independent groups have established the validity of the assay using even shorter time for systemic circulation time after tail vein injections including 10-minutes (PMID: 22523295; PMID: 29301791), 15 minutes (PMID: 32312382) or 30 minutes (PMID: 31866994). Therefore, the use of 20 minutes is very reasonable for systemic circulation provides enough circulation time to the reach the proximal retinal tissue. As noted in our previous rebuttal, this technique and duration of 20 minutes

have been used by our group & others ({Elshaer, 2019 #244; Mohamed, 2018 #245; Mysona, 2013 #246; Al-Shabrawey, 2008 #248; Antonetti, 1998 #249}.

Fig. 2A, the leukostasis has served the purpose of the experiment, but does not show whether the inflammatory cells have (or have not) migrated into the retinal tissue where they can exert their functions.

Retinal capillary cell death can occur either directly due to different biochemical insults initiated within retinal EC themselves, or indirectly secondary to the activation of non-retinal cell types; mainly circulating or infiltrating leukocytes {Kern, 2007 #137}. Retinal leukostasis is one of the established indirect prerequisite events for inducing exacerbated endothelial cell death and BRB breakdown, and its inhibition prevents retinal microvascular degeneration {Lutty, 2013 #185;Adamis, 2008 #138;Joussen, 2004 #136}. Therefore, retinal leukostasis as a well-established retinal microvascular assay, is a legitimate functional assay which represents the effective link between increased interaction between inflammatory and endothelial cells, (mediated by the increased endothelial pro-inflammatory adhesion molecules expressions, ex: ICAM-1 & VCAM-1), and the resulting physical occlusion of the poorly perfused retinal microvessels, which establishes the foundation for retinal micro-ischemia and increased acellular capillaries formation as shown in the current study results. The focus of the current study is to examine the role of HFD-induced inflammasome activation in mediating retinal microvascular dysfunction itself. Hence, it is beyond the scope of the current study to investigate the migration of the inflammatory cells into the extra-vascular retinal tissue (i.e: outside the retinal microvascular tissue).

  1. Mohamed, I.N., et al., Thioredoxin-interacting protein is required for endothelial NLRP3 inflammasome activation and cell death in a rat model of high-fat diet. Diabetologia, 2014. 57(2): p. 413-23.
  2. Abdelsaid, M.A., S. Matragoon, and A.B. El-Remessy, Thioredoxin-interacting protein expression is required for VEGF-mediated angiogenic signal in endothelial cells. Antioxid Redox Signal, 2013. 19(18): p. 2199-212.
  3. Elshaer, S.L., et al., Modulation of the p75 neurotrophin receptor using LM11A-31 prevents diabetes-induced retinal vascular permeability in mice via inhibition of inflammation and the RhoA kinase pathway. Diabetologia, 2019. 62(8): p. 1488-1500.
  4. Mohamed, R., et al., Inducible overexpression of endothelial proNGF as a mouse model to study microvascular dysfunction. Biochim Biophys Acta Mol Basis Dis, 2018. 1864(3): p. 746-757.
  5. Mysona, B.A., et al., Modulation of p75(NTR) prevents diabetes- and proNGF-induced retinal inflammation and blood-retina barrier breakdown in mice and rats. Diabetologia, 2013. 56(10): p. 2329-39.
  6. Al-Shabrawey, M., et al., Role of NADPH oxidase in retinal vascular inflammation. Invest Ophthalmol Vis Sci, 2008. 49(7): p. 3239-44.
  7. Antonetti, D.A., et al., Vascular permeability in experimental diabetes is associated with reduced endothelial occludin content: vascular endothelial growth factor decreases occludin in retinal endothelial cells. Penn State Retina Research Group. Diabetes, 1998. 47(12): p. 1953-9.
  8. Doczi-Keresztesi, Z., et al., Retinal and renal vascular permeability changes caused by stem cell stimulation in alloxan-induced diabetic rats, measured by extravasation of fluorescein. In Vivo, 2012. 26(3): p. 427-35.
  9. Kern, T.S., Contributions of inflammatory processes to the development of the early stages of diabetic retinopathy. Exp Diabetes Res, 2007. 2007: p. 95103.
  10. Lutty, G.A., Effects of diabetes on the eye. Invest Ophthalmol Vis Sci, 2013. 54(14): p. ORSF81-7.
  11. Adamis, A.P. and A.J. Berman, Immunological mechanisms in the pathogenesis of diabetic retinopathy. Semin Immunopathol, 2008. 30(2): p. 65-84.
  12. Joussen, A.M., et al., A central role for inflammation in the pathogenesis of diabetic retinopathy. FASEB J, 2004. 18(12): p. 1450-2.

Reviewer 3 Report

There is still no reference to Supplementary-Figure-4 in the paper.

I suppose the authors do not have a substantially better immunoblot for Figure S3 and one will have to rely on the quantification, which seems rather to represent an estimate given the quality of the blots. (The reason provided by the authors for the low quality seems irrelevant as the samples were derived from cultured cells).

Author Response

There is still no reference to Supplementary-Figure-4 in the paper.

We apologize for overlooking this mistake. Please see revised results section describing the findings of Supplementary-Figure-4.

I suppose the authors do not have a substantially better immunoblot for Figure S3 and one will have to rely on the quantification, which seems rather to represent an estimate given the quality of the blots. (The reason provided by the authors for the low quality seems irrelevant as the samples were derived from cultured cells).

We apologize for the unclear presentation of the contrast of the supplementary figures. We realized that there was probably some change that might have happened during manuscript preparation. Please see updated western blot representatives for the supplementary figures, which shows the consistent trend of significantly increased levels in the TXNIP overexpression group only versus other groups.

Round 3

Reviewer 3 Report

My comments have been adequately addressed